# Ex Vivo Perfusion of Porcine Pancreas and Liver Sourced from Commercial Abattoirs after Circulatory Death as a Research Resource: A Methodological Study

**DOI:** 10.3390/mps6040066

**Published:** 2023-07-12

**Authors:** Zainab L. Rai, Morenike Magbagbeola, Katie Doyle, Lukas Lindenroth, George Dwyer, Amir Gander, Agostino Stilli, Danail Stoyanov, Brian R. Davidson

**Affiliations:** 1Wellcome/EPSRC Centre for Interventional and Surgical Sciences (WEISS), University College London, London W1W 7TY, UK; 2Centre for Surgical Innovation, Organ Repair and Transplantation (CSIORT), University College London, London NW3 2PS, UK; 3Department of HPB & Liver Transplantation Surgery, Royal Free London NHS Foundation Trust, Pond Street, London NW3 2QG, UK; 4Department of Surgical & Interventional Engineering, School of Biomedical Engineering & Imaging Sciences, King’s College, London SE1 1UL, UK

**Keywords:** machine perfusion, whole-organ research, 3Rs

## Abstract

Background: Machine perfusion (MP) is increasingly used for human transplant organ preservation. The use of MP for research purposes is another opportunity for this technology. The porcine pancreas and liver are similar in anatomical size and function to their human counterparts, making them an excellent resource for research, but they have some important differences from human organs which can influence their research use. In this paper, we describe a technique developed and tested for the retrieval of porcine organs for use in research on perfused viable organs. Methods: Whole-organ porcine pancreata and livers were harvested at a commercial abattoir, following standard slaughterhouse processes. The standard slaughterhouse process involved a thoracotomy and mid-line laparotomy, and all the thoracoabdominal organs were removed. The pancreas, fixed in the retroperitoneum, was carefully dissected from its attachments to the surrounding structures, and tissue planes between the pancreas, spleen, duodenum, and colon were meticulously identified and dissected. Vessel exposure and division: The aorta, portal vein (PV), hepatic vein (HV), and hepatic artery (HA) were dissected and isolated, preserving the input and output channels for the liver and pancreas. A distal 3 cm of the aorta was preserved and divided and served as the input for the pancreas perfusions. The liver, PV, HV, and HA were preserved and divided to preserve the physiological channels of the input (PV and HA) and output (HV) for the liver perfusions. The porcine hepatic and pancreas anatomy shares significant resemblance with the gross anatomy found in humans, and this was taken into consideration when designing the perfusion circuitry. The porcine pancreas and spleen shared a common blood supply, with branches arising from the splenic artery. The organs were flushed with cold, heparinised normal saline and transported in a temperature-regulated receptacle maintained at a core temperature between 4 and 8 °C, in line with the standards of static cold storage (SCS), to a dedicated perfusion lab and perfused using our novel perfusion machine with autologous, heparinised porcine blood, also collected at the abattoir.

## 1. Introduction

Organ transplantation has revolutionised patient outcomes by increasing survival and ensuring a better quality of life since its inception in the 1950s [1,2]. Solid-organ transplantation has catalysed the development of new technologies in organ preservation and perfusion. In recent years, machine perfusion (MP) has been re-introduced for solid organ preservation in the hope of reducing organ preservation injury and possibly increasing the utilisation of marginal organs [3,4,5]. MP may also represent an opportunity for the research community to study new drugs, devices, and technologies on viable perfused organs. However, human organs are in short supply for both clinical transplantation and for research. An alternative is to use perfused animal organs for research.

The porcine pancreas and liver are similar in size and gross anatomy to their human counterparts [6] and may provide a potential source for experimental whole organs. According to data from the UK Department for Environment, Food and Rural Affairs (DEFRA), approximately 10 million pigs were slaughtered in the UK in 2020 [7], thus providing a potential source of organs for perfusion experiments.

Short-term studies of perfused porcine organs can contribute towards achieving several key research objectives. For example, the perfused animal organ research platform can assist in initial feasibility studies assessing new medicinal compound efficacy in the context of establishing the pharmacodynamics and pharmacokinetics of new medicinal compounds. Kassimatis et al. reported on a novel compound, Microcept, serving to delay reperfusion-ischaemia injury in perfused porcine kidneys and reported an optimal dose of 80 mg in perfused porcine kidneys [8]. Moreover, such a research platform may prove vital in the development of new methods of tissue ablation by establishing the optimal parameters of ablation in relation to tissue types and treatment zones. Finally, a perfusion circuit using animal organs may be used to assess novel technologies and devices for whole-organ blood flow dynamics.

The medical research community must also consider and adhere to the principles of the 3Rs (Replacement, Reduction and Refinement), developed more than half a century ago [9]. These principles offer a guiding structure to investigators for accomplishing animal research that is more humane and compassionate. Since its inception, the 3Rs have been incorporated into both national and international regulations relating to the use of animals in research and continue to enjoy the support of the wider public [10]. It is thus imperative that these principles are adhered to and respected. One way that this can be achieved is by utilising organs from a commercial abattoir in which the organs are routinely removed and discarded after the animal is terminated. Offal that is usually discarded can be used for some research purposes, as demonstrated here.

There is a need for detailed descriptions of the organ retrieval process from commercial sources. One paper reported the use of animal organs from a commercial slaughterhouse for perfusion. However, it failed to provide sufficiently detailed instructions to allow for the retrieval process [11]. The authors of that study reported on the procurement of porcine intra-abdominal organs from a commercial slaughterhouse and provided sufficient detail on the procurement of all intra-abdominal organs but failed to provide a detailed description on the crucial steps required for the transport and treatment of the organs and porcine blood with appropriate anticoagulation and specific anatomical landmarks for specific organs to assist in the retrieval process.

In contrast, this paper provides a comprehensive protocol for organ retrieval, including specific anticoagulation regimens and the identification of critical anatomical structures to ensure successful perfusion. By addressing these key gaps in the literature, our work provides an important contribution to the field of MP research using animal organs from commercial sources. Here, we describe the evolution in the retrieval technique adopted in the course of developing a novel organ perfusion system for the use in basic experimental research. The perfusion system and monitoring have been described elsewhere [12].

## 2. Experimental Design

### 2.1. Protocol for Organ Harvest at the Abattoir

#### 2.1.1. Selection of the Abattoir

A commercial abattoir located less than 50 miles from our dedicated perfusion lab was selected and agreed to take part in the study. This location was chosen due to its proximity to our lab, which allowed for the efficient transport of the organs and minimised the time from organ retrieval to the start of perfusion. By partnering with a nearby abattoir and implementing careful transportation protocols, we were able to obtain a steady supply of organs for our study while maintaining the quality and viability of the organs. The transportation process was carefully planned and optimised to ensure that the organs were handled appropriately and with minimal delay.

The commercial slaughterhouse from which the organs were obtained held a valid Certificate of Competence (CoC) from the UK Food Safety Agency (FSA), ensuring compliance with the routine animal hygiene standards set forth by the agency [13].

#### 2.1.2. Animal Selection

Domesticated sows and piglets (*Sus scrofa* Domesticus) were procured from a commercial. The animals were selected based on size, as larger animals (>80 kg) result in larger organs, requiring a greater volume of flush and anticoagulation. Therefore, the animals were selected based on age and size, i.e., the animals selected weighed between 50 kg and 80 kg with a maximum age of 12 months. The stunning and exsanguination of the animals are part of standard procedures. To minimise the warm ischemia time (WIT) and ensure timely retrieval of the organs, it was imperative that the initial workbench, where the organs were isolated and resected, was in close proximity to where the animals were slaughtered. Therefore, the abattoir kindly provided a workbench onsite for organ retrieval. The workbench was draped in sterile drapes, delineating a clean and easily identifiable area for the organ harvesters to work (Figure 1). To maintain strict hygiene standards, the organ retriever was not allowed to access the area where the animals were terminated.

Table 1 lists the essential equipment required in this stage of the retrieval process.

### 2.2. Materials and Animals

#### Standard Abattoir Procedure

The pigs were electrically stunned and exsanguinated through a vertical incision through the jugular vein and stab incision through the heart, which are standard procedures used in conventional slaughterhouses and consistent with FSA regulations. These standards include specific decontamination processes such as scalding. In the scalding process, which is an integral component of the meat processing industry, the carcasses of pigs are subjected to hot water to facilitate the loosening and removal of hair. As per the established procedures, the temperature of the water during scalding typically ranges between 60 and 63 degrees Celsius (140–145° Fahrenheit). It is noteworthy that the scalding process is primarily geared towards treating the exterior of the pig carcass and does not have an intended interaction with the internal organs. The skin and muscular layers of the carcass act as protective barriers, shielding the internal organs during the scalding process. As such, under standard operating conditions, there is no anticipated impact on the internal organs.

The examination of the animal organs was undertaken to ascertain the absence of conspicuous injuries. The protocol delineated herein is tailored to outline the procedure for retrieving animal organs with the objective of system development and preliminary investigations, rather than for transplantation purposes. Consequently, the stringent quality assessment measures that are typically associated with processes for organ transplantation are not deemed applicable in this context.

In this study, blood was carefully obtained from the animal by inverting it and facilitating the collection of blood into a receptacle pre-treated with heparin. The utilisation of hollow needles with vacuum assistance was deliberately circumvented as a method of blood collection to prevent potential harm to the blood components.

In the experiments requiring autologous blood used as a perfusate, blood was collected in a suitable receptacle containing unfractionated heparin sodium (Pfizer) to a ratio of 1 L of blood to 4000 I/U of heparin, a value arrived at as a result of trial and error during our protocol’s development. The previous collection and transport of the blood using values less than 4000 I/U resulted in coagulated blood, rendering it unusable in the machine perfusion, and anticoagulation greater than 4000 I/U resulted in highly anticoagulated blood. The anticoagulated blood was transported in a cold box, and the temperature was maintained between 5 and 10 °C for transport back to the perfusion lab. The blood temperature was measured at the start of transport using a standard scientific thermometer, upon arrival at the perfusion lab, and prior to perfusion.

## 3. Procedure



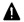

**CRITICAL STEP**
Perform a mid-line thoracotomy and laparotomy to provide access to all the organs of the thorax and abdominal compartments. This step is performed by the abattoir staff. Remove all the abdomino-thoracic organs en bloc. Place the contents of the thoraco-abdominal compartments, including the lungs, heart, aorta, and its branches, as well as the small and large bowel, on the dissecting table in the standard anatomical position to facilitate the easy identification of the organs (Figure 2). Once the liver has been identified and cleared of injuries and tears, the three main vessels, namely, the portal vein, hepatic vein, and hepatic artery, need to be identified. Identify and dissect a distal 1.5 cm of the hepatic vein.
Figure 2En masse retrieval of thoracic and abdominal organs, including the main vessels.
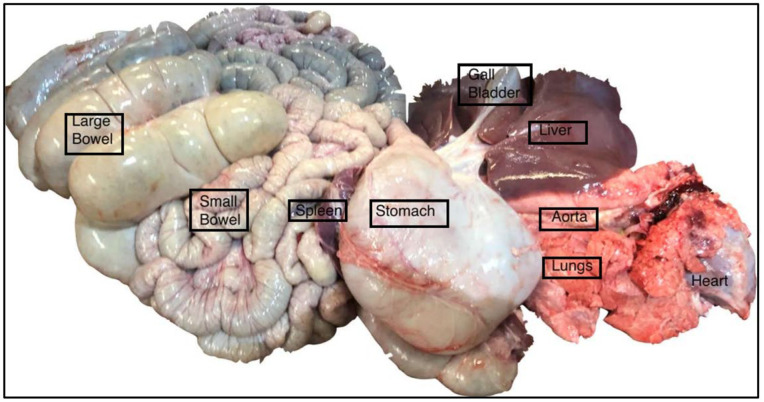
Inspect the liver for tears or injuries that would disqualify it from the next steps in the retrieval process. For instance, the liver in Figure 2 demonstrates a ragged edge and a retrieval injury and was subsequently discarded, demonstrating one of the advantages of using commercial abattoirs to source experimental organs.

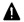

**CRITICAL STEP**
Ensure that the portal vein is dissected prior to dissecting the hepatic vein or artery to allow for a clear outflow for residual blood, reducing the chances of pooling and stagnation of the blood in the liver bed, leading to thrombi. Identify the hepatic artery and dissect it with a distal 2 cm (Figure 3). The hepatic artery can be identified by locating the gall bladder and using this as an initial anatomical reference point (see Figure 3). The gallbladder is connected to the common bile duct by the cystic duct, which can be traced from the gallbladder towards the place where it joins the common bile duct. As you locate the junction where the cystic duct meets the common bile duct, look for a blood vessel nearby that runs parallel to the common bile duct. This is the common hepatic artery. The common hepatic artery divides into the proper hepatic artery (which supplies the liver) and the gastroduodenal artery. To confirm that you have identified the hepatic artery, trace it towards the liver, where it further divides into the left and right branches supplying the respective lobes of the liver.
Figure 3Porcine liver demonstrating key anatomical references to help to identify important vasculature.
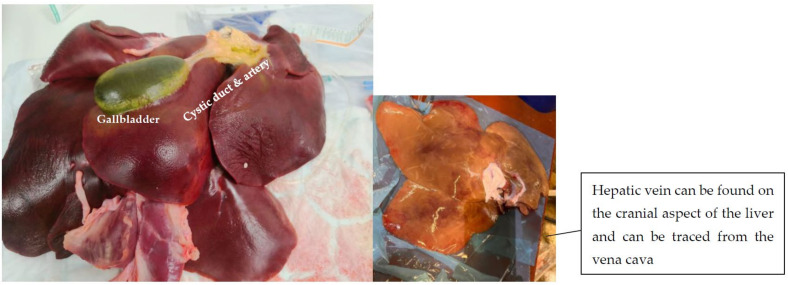
Once the hepatic artery has been identified and resected, flush 1 L of cold, heparinised normal saline through the hepatic artery using intermittent manual pressure. Finally, identify and resect the portal vein (Figure 4). Following the resection of all the surrounding hepatic ligaments and the completion of the heparinised flush, place the liver in a cooled receptacle maintaining a core temperature between 4 and 8 °C and transport to a dedicated perfusion lab.
Figure 4Liver dissection, demonstrating the identification and ligation of the hepatic artery and portal vein.
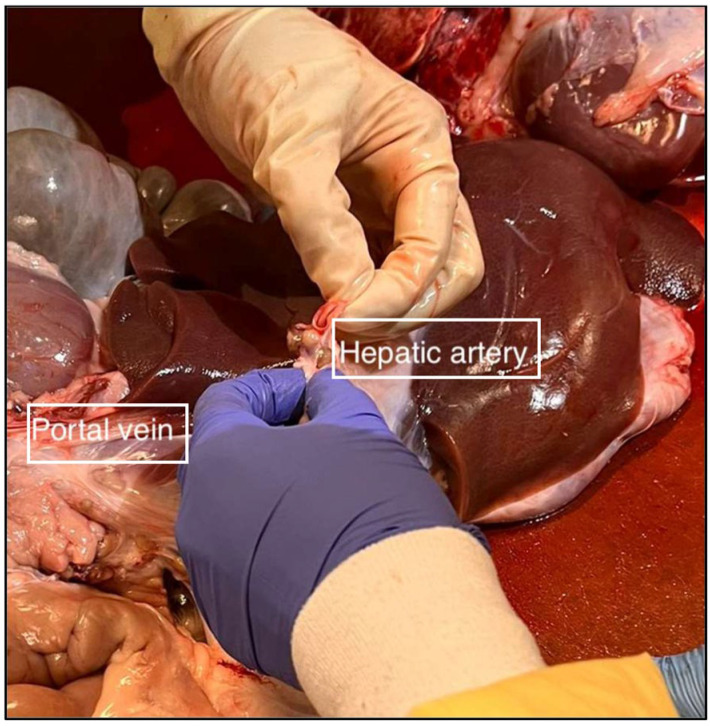


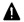

**CRITICAL STEP**
Identify the pancreas and inspect it for tears and inadvertent injury. The pancreas can be identified by locating the duodenum, following it distally from the stomach. The pancreatic head is located in the groove or C shape formed by the second and third parts of the duodenum (see Figure 4). Identify the aorta and follow it distally to the root of the coeliac axis. Using this anatomical point as a constant, identify a distal 3 cm of the aorta and incise it at this point. Immediately cannulate it with a soft 18 Fr T-tube attached to a giving set and flush a litre of cold, heparinised normal saline containing 50,000 I/U of streptokinase.

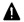

**CRITICAL STEP**
The porcine pancreas has a very thin and delicate capsule that can be damaged during the dissection process. However, this capsule is more robust around the pancreatic head. Therefore, taking this into account during the retrieval process, the following methodology focuses on the dissection of the pancreatic head alone (see Figure 5).
Figure 5Anatomical location of the pancreatic head in relation to the duodenum.
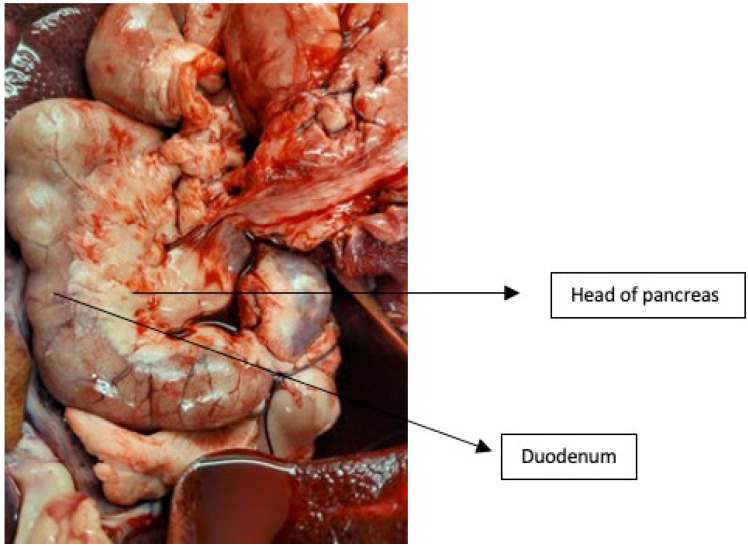

Free the large bowel from the duodenum and pancreas using sharp dissection while taking care to preserve the pancreatic capsule. Once the mesenteric root has been identified, suture it to minimize the leakage of the heparinized normal saline and added fibrinolytic. Suture the proximal duodenum and divide distal to the pylorus. Dissect the stomach using sharp dissection, cutting the hepatogastric ligament along the lesser curve. The spleen, which is attached to the pancreas, should be removed from the pancreatic tail without damaging the pancreas using a combination of blunt and sharp dissection. Identify the splenic vasculature and tie it separately before being dividing. The duodenum and pancreas need to be dissected carefully to obtain an intact pancreatic graft, which should be resected together with the duodenum to avoid damaging the pancreatic capsule.


## 4. Expected Results

Using our methodology for the retrieval and immediate management of porcine organs from a commercial abattoir and subsequent machine perfusion (MP), we anticipate the retrieval of viable porcine livers and pancreata suitable for perfusion using MP systems. Real-time organ function can be monitored during perfusion by measuring perfusate parameters such as the pH, lactate levels, and oxygen saturation and by conducting histological analysis of the organs after perfusion. Through these analyses, we expect to observe differences in organ function and viability based on factors such as warm ischemia time (WIT) and the specific anticoagulation protocol used during perfusion. We also anticipate that the optimised anticoagulation and oxygenation protocols used in our MP system will maintain the viability of the organs and promote their preservation for use as a potential research model.

In situations in which the protocol is not successful, we expect to observe reduced organ viability and function, indicated by elevated perfusate lactate levels, decreased pH, and altered histological profiles. Below, we describe the key learning points from the development of this protocol.

### 4.1. Technical Aspects of Organ Retrieval and Organ Injury

During the organ retrieval process, we used a combination of blunt and sharp dissection to free the large bowel from the duodenum and pancreas, taking care to preserve the thin and delicate pancreatic capsule. The spleen was removed from the pancreatic tail without damaging the pancreas, and the splenic vasculature was identified and tied separately before being divided. The duodenum and pancreas were dissected carefully to obtain an intact pancreatic graft, which was resected together with the duodenum to avoid damaging the pancreatic capsule.

However, we did encounter some organ injuries during the retrieval process. Specifically, we observed damage to the pancreatic capsule in a small number of cases. We also observed some damage to the splenic vasculature during dissection, which required additional suturing to achieve haemostasis.

### 4.2. Developments in Retrieval Technique

Our protocol for pancreatic retrieval involved perfusing the pancreatic head alone, which has a more robust pancreatic capsule in the porcine anatomy and a more easily accessible blood supply via the aorta. This approach allowed us to minimize damage to the delicate pancreatic capsule and improve the quality of the grafts obtained.

### 4.3. Reasons for Not Using Commercial Organ Preservation Solutions and the Implications

We chose not to use commercial organ preservation solutions due to their high cost and limited availability. Instead, we used autologous blood with the addition of heparin and streptokinase to minimise the formation of thrombi affecting the perfusion circuit and the perfused organ. Additionally, the organs were perfused with heparinised saline and, in the case of porcine pancreata, the addition of streptokinase to reduce the formation of microthrombi in the microcirculation of the head of the pancreas. Our results suggest that this approach was effective in preserving organ viability and minimising damage during transport.

It is important to note that the use of heparin and streptokinase requires careful the monitoring and adjustment of the dosage to ensure that the organs are not over- or under-perfused, which can lead to tissue damage and/or graft failure. To this end, our protocol recommends using 50,000 I/U of streptokinase to flush the pancreatic head and 8000 I/U of heparin for flushing both the liver and the pancreas.

### 4.4. Attachment to MP and Technical Issues

During organ perfusion, we attached the organs to a centrifugal pump to maintain a consistent perfusion pressure and flow. However, we did encounter any technical issues during attachment, including air bubbles in the tubing and difficulty maintaining consistent flow rates. These issues were addressed through careful monitoring and adjustment of the pump settings.

### 4.5. Demonstration of Uniform Perfusion and Duration

We demonstrated uniform perfusion and duration across all the pancreatic grafts using a combination of macroscopic examination and histological analysis. Studies using spectral imaging and simultaneous imaging during perfusion to validate uniform perfusion are currently underway; however, the preliminary results from these studies are promising and corroborate the histological data.

### 4.6. Measures of Viability and Time Frame

We measured organ viability using a combination of metabolic and histological assays. Specifically, we measured ATP levels and bile production as indicators of metabolic activity and general measures of organ functionality in the liver perfusions. For both the liver and pancreas perfusions, organ weight, perfusate pH, and histological evidence of tissue damage were measured as indicators of organ viability.

All organs were perfused and transported within 2 h of retrieval, and graft connection to the MP circuit was performed within 3 h of retrieval to ensure maximal organ viability.

By providing an alternative investigation model using porcine organs perfused ex vivo, we aim to contribute to the development of a novel investigative platform that utilises an MP system that is versatile enough to perfuse several organs and contribute to the development of new pharmacological agents and technologies. Our methodology has the potential to provide valuable insights into the optimal conditions for the perfusion of abattoir organs and improve the viability of organs for transplantation.

## 5. Conclusions

Based on the findings presented in this paper, we successfully developed and tested a methodology for the retrieval and perfusion of porcine pancreata and livers using a novel perfusion machine. Our results demonstrate that this technique is both feasible and effective for perfusing whole ex vivo organs and can be used as an alternative investigation model for research purposes. Further studies are needed to explore the potential applications of this methodology in the field of organ preservation and transplantation.

## Figures and Tables

**Figure 1 mps-06-00066-f001:**
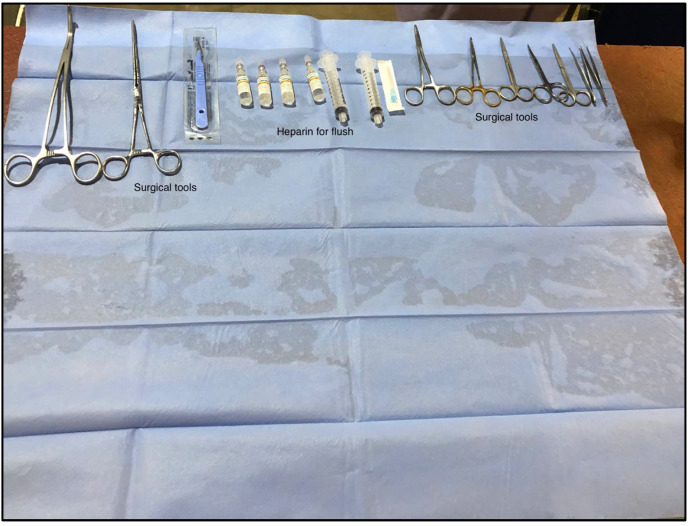
Draped workbench onsite at the local abattoir and surgical tools required to perform organ retrieval.

**Table 1 mps-06-00066-t001:** Abattoir equipment list.

Equipment	Purpose	Quantity	Disinfection Procedure
Personal Protective Equipment	To protect workers from exposure to blood and other bodily fluids	N/A	N/A
Drapes for workbench	To provide a clean and sterile work surface	1/procedure	N/A
Surgical equipment	To dissect and isolate the organs	As needed	Sterilize with ethylene oxide gas or steam
Heparinised normal saline	To perfuse the organs during retrieval	2 L/procedure	N/A
Drip stand	To support the bag of heparinised saline	1/procedure	Clean with disinfectant solution

## Data Availability

Data from this study is currently being analysed and not publicly available due to privacy restrictions.

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
