# Peer review of "Ex Vivo Perfusion of Porcine Pancreas and Liver Sourced from Commercial Abattoirs after Circulatory Death as a Research Resource: A Methodological Study"

_mps, 2023, doi:10.3390/mps6040066_

Round 1
Reviewer 1 Report
The study "Ex Vivo Perfusion of Porcine Pancreas and Liver Sourced from 2 Commercial Abattoirs after Circulatory Death as a Research Re- 3 source: A Methodological Study on Technique" by Rai et al is presented for peer review. The authors demonstrate a technique for procuring slaughterhouse organs for experimental machine perfusion.
While the study is of general interest its lacks an in-depth visual description of the suggested procedure.
There is an urgent need for more and especially more detailed pictures of the anatomy. How is the common hepatic artery identified? How is the mesenteric rout cut? How to identify the pancreas? How to perfuse the pancreas? All that has to bee shown in pictures.
NA
Author Response
Point 1
Need for more and especially more detailed pictures of the anatomy.
The authors have added extra pictures demonstrating key anatomical references and linking this back to the text to add further clarification and detail to the retreival protocol.
Point 2
How is the common hepatic artery identified?
Please refer to the additional pictures (Figure 3) which show photographically the anatomical references used to identify the hepatic artery. This is explained in additional text to add further clarification
Point 3
How is the mesenteric rout cut?
The mesenteric root is not cut, instead it is sutured to ensure that the anticoagulation admisistered remains within the organ ensuring thourough anticoagulation. This has been explained in the the protocol under section 7.
Point 4
How to identify the pancreas?
An additional picture (Figure 4) has been added that shows the location of the pancreas and further text has also been added as an adjunt to the figure to provide further clarification on how to identify the pancreas.
Point 5
How to perfuse the pancreas? All that has to bee shown in pictures.
The authors feel that detailed description on how the pancreas is perfused is beyond the scope of this article. The protool submitted describes the retreival process and immediate management of organs retreived from a commercial abattoir, as is appropriate for an article for the Methods and Protocol. Detailed explanation of perfusion and results has been published and referenced in the text (please see Magbagbeola M, Rai ZL, Doyle K, Lindenroth L, Dwyer G, Gander A, et al. An adaptable research platform for ex vivo normothermic machine perfusion of the liver. International journal of computer assisted radiology and surgery. 2023.)
Reviewer 2 Report
The authors describe a new and safe method for retrieving pig organs from slaughterhouse donors. This method is considered in terms of the 3Rs (Refinement, Reduction, and Replacement) and has the potential to reduce the number of animals used. However, the description of certain points is quite superficial, and limitations of this potential methodology are not discussed. Consequently, several questions regarding the technique remain unanswered:
- The blood is collected after the bleeding cut (from the heart or carotid artery), and in most cases in slaughterhouses, hollow knives with vacuum systems are used, which destroy the blood cells due to the high pressure. Which method of bleeding was applied in this case? How was it ensured that the blood components remained suitable as a perfusate?
- The authors state that after stunning and bleeding the animals, the organs were immediately removed. In some European countries, after bleeding the animals, it is mandatory to perform carcass cleaning, hair removal, and disinfection using scalding methods. The intact carcasses are scalded for a few minutes at 60-63 degrees. Was this also done? How does this heat treatment affect the removal process and the quality of the organs?
What measures are in place to ensure the quality of the retrieved organs? Are there any protocols for testing and evaluating the organs before transplantation?
In the abstract, laboratory investigations, ischemia time, AST values, and production values (bile) were described, but they are neither listed nor shown within the paper. To assess this method, the vitality of the organs should be demonstrated based on data.
Overall, this paper lacks structured data collection to evaluate the effectiveness of the technique being touted. Additionally, providing graphical or photographic representations of the critical steps involved in the pancreas preparation would be beneficial. However, the schematic representation 4a and 4b, in particular, do not add any value.
Overall, this paper has potential, as resources for investigating pig organs within transplantation research are limited, and there is still a need for reduction in line with the 3Rs principles. However, the investigations conducted here are not sufficient to evaluate the effectiveness of this methodology. Detailed descriptions of pancreas preparation and perfusion, which are not trivial in pigs, are also lacking. Furthermore, the dimensions and weights of the pigs were not clearly described. The weights and respective ages of the organs are particularly important in transplantation research due to the significant differences in vascular quality at different stages of maturity. These aspects should be addressed and emphasized in the paper.
.
Author Response
Point 1
Which method of bleeding was applied in this case? How was it ensured that the blood components remained suitable as a perfusate?
A detailed description of how the blood was collected has been added to provide the clarification requested.
Point 2
Was this [scalding post stunning] also done? How does this heat treatment affect the removal process and the quality of the organs?
Scalding was implemented as per standard U.K FSA procedures. The process of scalding is specifically to facilitate hair removal and overall surface hygiene and no effect was observed with regards to the quality of the internal organs due to the thick muscle and fat layer surrounding the internal organs. Additional text has been added to make and clarify this.
Point 3
What measures are in place to ensure the quality of the retrieved organs? Are there any protocols for testing and evaluating the organs before transplantation?
The authors have made the point clearer in the manuscript that the purpose of these retreived organs is not for transplantation but as a potential investigative platform using machine perfusion technology. As such, stringent quality checks performed in the clinical setting or where the organs are destined for transplantation are not appropriate nor required.
Point 4
To assess this method, the vitality of the organs should be demonstrated based on data.
The authors would like to kindly clarify that the focus of the manuscript being submitted to this journal is to describe the methodology involved in the retrieval and immediate management of organs from a commercial abattoir. As per the journal’s template and guidelines for methodological papers, a results section is not a requirement for submission. Furthermore, the results pertaining to the assessment of organ vitality, which you mentioned, have already been published elsewhere (see Magbagbeola M, Rai ZL, Doyle K, Lindenroth L, Dwyer G, Gander A, et al. An adaptable research platform for ex vivo normothermic machine perfusion of the liver. International journal of computer assisted radiology and surgery. 2023). This includes the data-based demonstration of organ vitality, which is beyond the scope of the current methodological manuscript.
Point 5
Detailed descriptions of pancreas preparation and perfusion, which are not trivial in pigs, are also lacking.
Please see response to Point 4
Dimensions and weights of the pigs were not clearly described. These aspects should be addressed and emphasized in the paper.
Additional text has been added to emphasize the weights of the pigs
Round 2
Reviewer 1 Report
The authors have significantly improved the manuscript.